biochemistry

flavonoid glycosyltransferases, *Andrographis paniculata*, substrate spectrum

**Authors for correspondence:**
Wei Gao
e-mail: weigao@ccmu.edu.cn
Lu-Qi Huang
e-mail: huangluqi01@126.com

# Functional characterization of three flavonoid glycosyltransferases from *Andrographis paniculata*

Yuan Li[1,2], Xin-Lin Li[4], Chang-Jiang-Sheng Lai[2],
Rui-Shan Wang[2], Li-Ping Kang[2], Ting Ma[1],
Zhen-Hua Zhao[1], Wei Gao[3,4,5] and Lu-Qi Huang[1,2]

[1]Shandong University of Traditional Chinese Medicine, Jinan 250355, People's Republic of China
[2]National Resource Center for Chinese Materia Medica, China Academy of Chinese Medical Sciences, Beijing 100700, People's Republic of China
[3]School of Pharmaceutical Sciences, [4]School of Traditional Chinese Medicine, and [5]Advanced Innovation Center for Human Brain Protection, Capital Medical University, Beijing 100069, People's Republic of China

(iD) YL, 0000-0001-9942-0803; WG, 0000-0003-3081-9642

*Andrographis paniculata* is an important traditional medicinal herb in South and Southeast Asian countries with diverse pharmacological activities that contains various flavonoids and flavonoid glycosides. Glycosylation can transform aglycones into more stable, biologically active and structurally diverse glycosides. Here, we report three glycosyltransferases from the leaves of *A. paniculata* (ApUFGTs) that presented wide substrate spectra for flavonoid glycosylation and exhibited multi-site glycosylation on the substrate molecules. They acted on the 7-OH position of the A ring and were able to glycosylate several other different types of compounds. The biochemical properties and phylogenetic analysis of these glycosyltransferases were also investigated. This study provides a basis for further research on the cloning of genes involved in glycosylation from *A. paniculata* and offers opportunities for enhancing flavonoid glycoside production in heterologous hosts. These enzymes are expected to become effective tools for drug discovery and for the biosynthesis of derivatives via flavonoid glycosylation.

## 1. Introduction

*Andrographis paniculata* has various pharmacological properties, including anti-inflammatory [1], antihyperglycaemic [2,3], hepatoprotective [4,5], anti-cancer [6,7], antihyperlipidaemic

[7,8], antioxidant [9,10], antimicrobial [11–13] and antiparasitic activities [14]. It is one of the most commonly used traditional medicinal herbs in South and Southeast Asian countries and has great potential for further applications [15–17]. Flavonoids and their glycosides are among the predominant secondary metabolites in *A. paniculata* and have a basic benzopyran ring nucleus skeleton formed by a part of the phenylpropanoid metabolism network [18–24]. Flavonoids in the form of glycosides play pivotal roles in the growth and development of plants by regulating the homeostasis of auxin hormones [25,26]. In recent years, increasing attention has been paid to the pharmacological activities of flavonoid glycosides from *A. paniculata* including antiplatelet and antiproliferative activities, which offered opportunities for further development and clinical application of this herb [15–17]. Glycosylation is the key modification step in various biological processes, especially in secondary metabolic pathways. It changes the stability, polarity, solubility, bioactivity, toxicity and subcellular localization of the substrate molecules [27–32]. Great progress has been made in chemical and enzymatic glycosylation in recent decades. However, the chemical glycosylation reactions have some limitations, such as redundant side reactions and intermediates, poor regio- and stereoselectivities, low yields, limited solvent compatibility, complicated extraction and separation as well as tedious protection–deprotection steps [33–36].

The glycosylation of both natural and unnatural products by glycosyltransferases, which is a new field of synthetic glycobiology, is more efficient in the production of glycosides than chemical approaches and has developed quickly in recent years [37–46]. The discovery of novel glycosyltransferases is of great value to the elucidation and prediction of glycoside biosynthetic pathways [29]. Glycosylation is the key modification step in various biological processes that produce many natural products containing diverse sugar moieties and increase drug availability. The enzymes that catalyse glycosylation reactions belong to the glycosyltransferase superfamily. Glycosyltransferases (EC 2.4.x.y) catalyse the transfer of sugar moieties from activated donor molecules to a wide range of acceptor molecules, such as sugars, lipids, proteins, nucleic acids, antibiotics and other small molecules, including plant secondary metabolites [47].

As of January 2019, 106 families of glycosyltransferases could be found in the Carbohydrate-Active Enzymes Database (CAZy) (http://www.cazy.org/GlycosylTransferases.html). Among those families, family 1 glycosyltransferases (GT1s) is the largest family in the plant kingdom [48]. GT1s are often referred to as UGTs because they typically transfer a sugar residue from UDP-glucose donors to specific acceptor molecules. UGTs contain a conserved PSPG (plant secondary product glycosyltransferase) box in the C-terminus protein domain. It consists of 44 amino acid residues and functions as a nucleoside-diphosphate-sugar binding site of the enzymes [49]. With the exception of the PSPG domain, UGTs share relatively low sequence identity. However, their secondary and tertiary structures are usually highly conserved. All these UGTs contain a GT-B fold, consisting of two separate Rossmann domains with a connecting linker, where the activated donor binds to the C-terminal domain and the acceptor binds to the N-terminal domain [50].

At present, few specific studies on flavonoid UDP-glycosyltransferases in *A. paniculata* (ApUFGTs) have been reported. We performed time-coursed transcriptome sequencing with MeJA (methyl jasmonate) treatment, three UGTs were identified to be capable of preferentially introducing a glucose on the 7-OH group of flavonoids as well as catalysing the glycosylation of flavones, isoflavones, flavanones, flavonols, dihydrochalcones and other small molecular aromatic compounds. The biochemical properties and phylogenetic analysis of ApUFGTs were also explored.

# 2. Material and methods

## 2.1. Chemicals and plant materials

Chemicals and reagents were purchased from Sigma-Aldrich (St Louis, MO, USA), J & K Scientific Ltd (Beijing, China), Chengdu Biopurify Phytochemicals Ltd (Chengdu, China) and BioBioPha (Kunming, China). *Andrographis paniculata* seeds were purchased from Zhangzhou, Fujian Province, China. The seeds were sterilized in 20% sodium hypochlorite solution containing 0.1% Triton X-100d for 10 min, washed five times with sterilized water and seeded on MS medium containing 0.7% agar. Uniformly sized two-week-old seedlings were supported on an adjustable plate and transferred to containers filled with 1 l Hoagland solution (pH 6.0), and grown in a controlled environment chamber, maintained at 25 ($\pm 2^\circ$C) under a 16/8 h (bright/dark) light cycle.

**Table 1.** Primer sequences used for cloning the full-length gene of ApUFGTs.

| name | sequence (5′–3′) |
| --- | --- |
| ApUFGT1-F | TCCAGGGGCCCGAATTCGGAATGGAGAATAATAACAAAGTTG |
| ApUFGT1-R | AGTGCGGCCGCAAGCTTGTTAGCTATATTTTTGTTGTAT |
| ApUFGT2-F | TCCAGGGGCCCGAATTCGGAATGTCGGCCGCCACCGCC |
| ApUFGT2-R | AGTGCGGCCGCAAGCTTGTTATTGTAACGATACAGCTC |
| ApUFGT3-F | TCCAGGGGCCCGAATTCGGAATGGATCCCAATGTCGAAG |
| ApUFGT3-R | AGTGCGGCCGCAAGCTTGTTACTTTGCTTCATTTTTCTC |

## 2.2. cDNA synthesis and gene cloning

UGTs were screened from *A. paniculata* transcriptome databases. To clone permissive ApUFGTs from *A. paniculata*, leaves of *A. paniculata* were treated with MeJA for 48 h prior to RNA isolation. The extracted RNA (Thermo Fisher Scientific, CA, USA) was used to synthesize cDNA using a PrimerScript$^{TM}$ RT Reagent Kit with gDNA Eraser (Takara, Dalian, China) according to the manufacturer's protocol. Full-length coding sequences of the selected UGTs were amplified by PCR using specific primers designed by Primer Premier 5.0 software (table 1). PCR was performed in a 100 µl scale using KOD-Plus-Neo (TOYOBO, Japan) at 94°C for 2 min; 35 cycles of 98°C for 10 s, annealing at 55°C for 30 s and extension at 68°C for 1 min; a final extension at 68°C for 5 min. PCR products were purified using a GeneJET Gel Extraction Kit (Thermo Scientific, USA) and ligated into the N-terminal MBP fusion expression vector HIS-MBP-pET28a (provided by Dr Xiaohong Zhang; HIS, histidine; MBP, maltose-binding protein) that had previously been digested with the restriction enzymes BamHI and SalI according to the protocol accompanying the pEASY-Uni Seamless Cloning and Assembly Kit (TransGen Biotech). The conjugates were transformed into Trans1 T1 phage-resistant chemically competent cells (TransGen Biotech, Beijing, China). The recombinant plasmids were obtained by screening positive clones and sequencing.

## 2.3. Sequence alignment and phylogenetic analysis

DNAMAN software was used to carry out the multiple alignment. ClustalW analysis software was used to compare the amino acid sequences of the flavonoid glycosyltransferases from other plant sources, and a phylogenetic tree was constructed using MEGA 7.0 software. Branch support was evaluated using bootstrap analysis with 1000 replicates [51].

## 2.4. Heterologous expression and affinity purification of ApUFGTs

The recombinant plasmids were transformed into *Escherichia coli* Transetta (DE3) expressing competent cells (TransGen Biotech, Beijing, China). The monoclonal colonies were identified and transferred to Luria–Bertani (LB) medium containing kanamycin (50 µg ml$^{-1}$). When the density of the host bacteria (OD$_{600}$) reached 0.6–1.0 following incubation at 37°C, an appropriate IPTG inducer (final concentration of approx. 1 mM) was added to induce culturing at a low temperature (16°C) for 12 h. The samples were subjected to centrifugation at 4°C for 20 min, suspension in lysis buffer (50 mM PBS (pH 7.4), 1 mM EDTA, 10% glycerol and 1 mM PMSF), disrupted by sonication in an ice bath (ultrasonic power 5 s, interval 5 s, continuous for 10 min) and followed by centrifugation at 10 000$g$ for 10 min. The crude proteins were filtered through a 0.45 µm membrane, transferred to an Ni-NTA agarose affinity column (Qiagen, WI, USA) and rotated at 4°C for 2 h to allow the Ni-NTA to fully bind to the protein. The samples were eluted with different concentrations of imidazole/PB buffer (0.02 M Na$_2$HPO$_4$–NaH$_2$PO$_4$ (pH 7.4) and 0.5 M NaCl with imidazole concentrations of 50, 100, 200, 300 and 500 mM). The proteins were then concentrated by Amicon Ultra-30 K filters (Millipore, USA), and finally, the buffer was changed to desalting buffer (50 mM Tris–HCl, pH 7.4). The protein concentrations were determined using a modified Bradford protein assay kit (Sangon Biotech, Shanghai, China), and the purified proteins were validated by SDS–PAGE.

## 2.5. Enzyme assays

The reaction system for the ApUFGT activity assay was as follows: a total volume of 100 µl containing 50 mM Tris–HCl (pH = 8.0), 8 µg purified proteins, 320 µM aglycone and 3200 µM UDP-glucose.

**Table 2.** UPLC methods used in this study.

| method | solvent A | solvent B | flow rate | gradient | analysis substrates |
|---|---|---|---|---|---|
| A | 0.1% formic acid | $CH_3CN$ | 0.4 ml min$^{-1}$ | 95–83% A (0–3 min), 83–65% A (3–12 min), 65–40% A (12–14.5 min) | **1–3, 12–14** |
| B | 0.1% formic acid | $CH_3CN$ | 0.4 ml min$^{-1}$ | 95–75% A (0–6 min), 75–60% A (6–15 min) | **4–11** |

The reaction was conducted at 40°C for 6 h and twice the volume of methanol was added to terminate the reaction; the mixture was shaken well, centrifuged at 12 000$g$ for 10 min, and then the supernatant was filtered through a 0.22 µm filter and subsequently analysed. The chromatographic analyses were conducted using a Waters Acquity UPLC-I-Class system (Waters Corp., Milford, MA, USA) with an Acquity UPLC BEH C18 column (1.7 µm, 2.1 × 50 mm). Gradient programmes were used to analyse the reaction mixtures (table 2). The PDA (photo-diode array) scanned from 190 to 400 nm. The total conversion rate was calculated to be 1% of the sum of the peak areas of the substrate and product(s). The glycosylated products were separated on a Waters UPLC system coupled with a Xevo G2-S QTOF-MS (Waters Micromass, Manchester, UK) with an Acquity BEH C18 column (50 × 2.1 mm, 1.8 µm). The following Q-TOF-MS parameters were used: ESI (+) ionization mode; scan range, 50–1500 Da; scan time, 0.2 s; cone voltage, 40 V; source temperature, 100°C; dissolved gas temperature, 450°C; cone gas flow rate, 50 l h$^{-1}$; desolvation flow rate, 900 l h$^{-1}$; and collision energy, 20–50 V. The mass accuracy was corrected by a lock spray with leucine enkephalin (200 pg µl$^{-1}$, 10 µl min$^{-1}$) as the reference ($m/z$ 556.2766 ESI (+)). Data were analysed using MassLynx$^{TM}$ software (v. 4.1, Waters Co., Milford, MA, USA).

## 2.6. Effects of temperature and pH on enzyme activities

The assays of the biochemical properties of temperature and pH were performed by changing each of the reaction conditions. To determine the optimal reaction temperature, the reaction mixtures were incubated at different temperatures (20, 30, 40, 50 and 60°C). To study the optimal pH, the enzymatic reactions were performed in various reaction buffers with pH values in the range of 4.0–11.0 (pH 4.0–7.0, citric acid–sodium citrate buffer; pH 7.0–9.0, Tris–HCl buffer; and pH 9.0–11.0, Na$_2$CO$_3$–NaHCO$_3$ buffer). All experiments were performed with UDPG as the donor and wogonin (10) as the acceptor in a total volume of 100 µl as described above. All experiments were carried out in triplicate. The mixtures were analysed by UPLC analysis as described in table 2. The total conversion rate was calculated to be 1% of the total peak area of the substrate and product.

# 3. Results

## 3.1. cDNA cloning of ApUFGTs

The three ApUFGTs, namely, ApUFGT1 (GenBank accession MH379334), ApUFGT2 (GenBank accession MH379339) and ApUFGT3 (GenBank accession MH379336), were deduced to code for a 485-amino acid protein (Mw: 54.669 kDa; pI: 5.03), a 479-amino acid protein (Mw: 51.561 kDa; pI: 6.20) and a 463-amino acid protein (Mw: 52.290 kDa; pI: 5.26), respectively (figure 1). A BLASTP procedure was used to find homologous genes with ApUFGTs, and these UGTs were aligned using DNAMAN software. ApUFGT1 has a high homology with a UGT from *Olea europaea* (GenBank accession XP_022868976) and with a UGT from *Strobilanthes cusia* (GenBank accession AZL90047). These genes all belong to the UGT86 family, and their homologies with ApUFGT1 were 53% and 52%, respectively. In addition, ApUFGT1 showed 50% homology with a UDP-glycosyltransferase from *Prunus yedoensis* (GenBank accession PQQ03238). ApUFGT2 has high homology with a UDP-glucuronosyl and a UDP-glucosyl transferase from *Handroanthus impetiginosus* (GenBank accession PIN09068) and the homologies were 66%. A flavonol 3-O-glucosyltransferase from *Cicer arietinum* (GenBank accession XP_004516861) and a UDP-glycosyltransferase

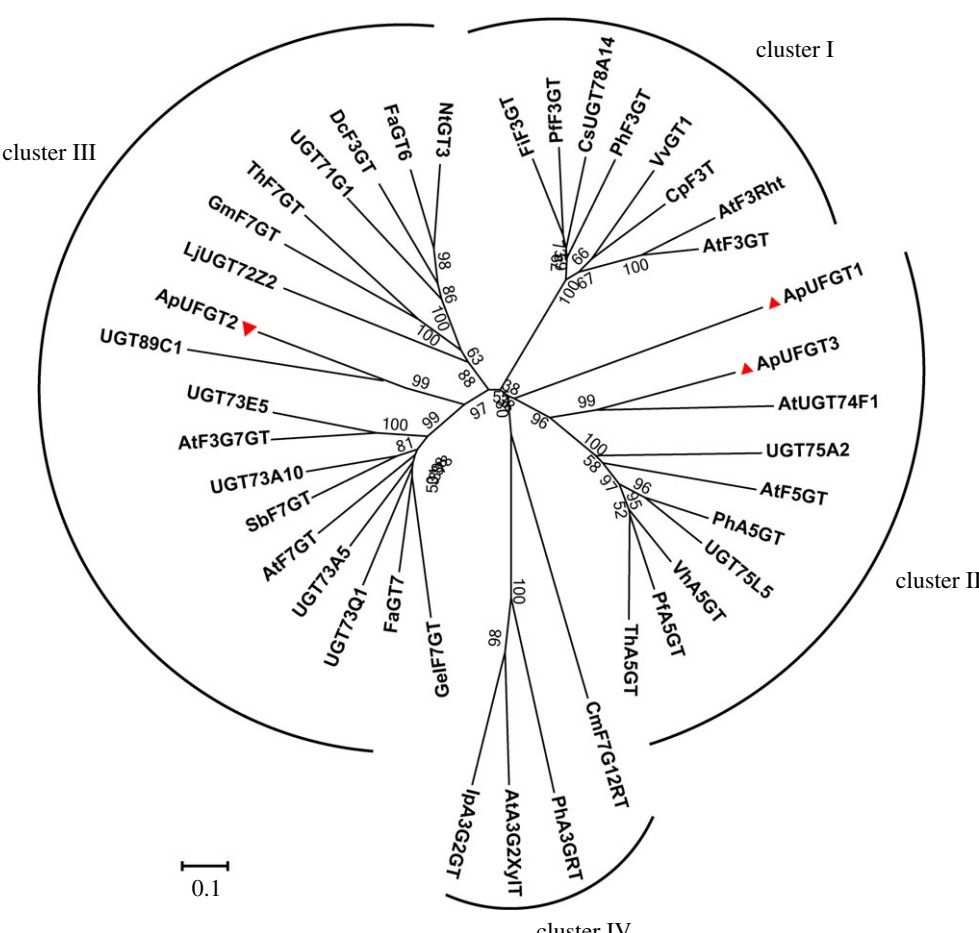

**Figure 1.** SDS – PAGE of recombinant ApUFGTs purified by affinity chromatography. M, standard protein markers (Thermo Scientific). 1, ApUFGT1; 2, ApUFGT3; 3, ApUFGT2.

**Figure 2.** Phylogenetic tree of ApUFGTs and other plant UGTs. The tree was constructed using MEGA 7.0 software with a 1000 bootstrap value. Clusters (I, II, III and IV) are shown in boldface letters. ApUFGT1, ApUFGT2 and ApUFGT3 are shown with red triangles. All the GenBank accession numbers of the sequences used in the phylogenetic analysis are indicated in table 3.

from *O. europaea* (GenBank accession XP_022869837) also showed high homologies with ApUFGT2; these genes belong to the UGT89 family, and their homologies were 51% and 63%, respectively. ApUFGT3 showed a homology of 52% with a UDP-glycosyltransferase from *Lycium barbarum* (BAG80541). In addition, ApUFGT3 has a high homology with a UGT from *Citrus clementina* (GenBank accession XP_006447932) and a UGT from *Morus notabilis* (GenBank accession XP_010095580), both of which belong to the UGT74 family. Their homologies with ApUFGT3 were both 51%.

## 3.2. Phylogenetic and sequence analysis of ApUFGTs

A phylogenetic tree was constructed using MEGA 7.0 by a neighbour-joining distance analysis based on the deduced amino acid sequences of the three ApUFGTs and other flavonoid glycosyltransferases downloaded from NCBI (https://www.ncbi.nlm.nih.Gov/) (figure 2 and table 3). ApUFGT2 was clustered with the 17 other UFGTs in cluster III. ApUFG1 and ApUFG3 were both clustered with the

**Table 3.** Sequences information used in phylogenetic tree in figure 2.

| gene name | accession/number | species | function |
|---|---|---|---|
| UGT89C1 | AAP31923 | *Arabidopsis thaliana* | flavonol 7-O-rhamnosyltransferase |
| AtF7GT | AKQ76388 | *Arabidopsis thaliana* | flavonoid 7-O-glucosyltransferase |
| UGT73A5 | CAB56231 | *Dorotheanthus bellidiformis* | betanidin 5-O-glucosyltransferase |
| DcF3GT | BAD52004 | *Dianthus caryophyllus* | flavonol 3-O-glucosyltransferase |
| FiF3GT | AAD21086 | *Forsythia intermedia* | flavonoid 3-O glucosyltransferase |
| GeIF7GT | BAC78438 | *Glycyrrhiza echinata* | isoflavonoid 7-O-glucosyltransferase |
| GmF7GT | NP001235161 | *Glycine max* | isoflavonoid 7-O-glucosyltransferase |
| PfF3GT | BAA19659 | *Perilla frutescens* | flavonoid 3-O-glucosyltransferase |
| PfA5GT | BAA36421 | *Perilla frutescens* var. crispa | anthocyanin 5-O-glucosyltransferase |
| PhF3GT | BAA89008 | *Petunia hybrida* | anthocyanin 3-O-glucosyltransferase |
| PhA5GT | BAA89009 | *Medicago truncatula* | anthocyanin 5-O-glucosyltransferase |
| PhA3GRT | CAA50376 | *Petunia hybrida* | anthocyanidin 3-O-glycoside rhamnosyltransferase |
| SbF7GT | BAA83484 | *Scutellaria baicalensis* | flavonoid 7-O-glucosyltransferase |
| ThA5GT | BAC54093 | *Torenia hybrida* | anthocyanin 5-O-glucosyltransferase |
| VhA5GT | BAA36423 | *Verbena hybrida* | anthocyanin 5-O-glucosyltransferase |
| VvGT1 | AAB81682 | *Vitis vinifera* | flavonoid 3-O-glucosyltransferase |
| AtF3Rht | AAM65321 | *Arabidopsis thaliana* | flavonoid 3-O-glucosyltransferase |
| CpF3T | ACS15351 | *Citrus paradise* | flavonoid 3-O-glucosyltransferase |
| CsUGT78A14 | A L019888 | *Camellia sinensis* | flavonoid 3-O-glucosyltransferase |
| AtUGT74F1 | NP973682 | *Arabidopsis thaliana* | UDP-glycosyltransferase 74 F1 |
| FaGT7 | Q2V6J9 | *Fragaria ananassa* | flavonoid 3-O-glucosyltransferase |
| AtF3G7GT | Q9ZQ95 | *Arabidopsis thaliana* | flavonol-3-O-glycoside-7-O-glucosyltransferase |
| LjUGT72Z2 | KP410264 | *Lotus japonicus* | flavonoid glycosyltransferase |
| GmF7GT | NP001235161 | *Glycine max* | isoflavone 7-O-glucosyltransferase |
| FaGT6 | Q2V6K0 | *Fragaria ananassa* | flavonoid 3-O-glucosyltransferase |
| CmF7G12RT | AAL06646 | *Citrus maxima* | flavonoid 1 – 2 rhamnosyltransferase |
| AtA3G2XylT | NP200217 | *Arabidopsis thaliana* | flavonoid 3-O-glucosyltransferase |
| CmF7G12RT | AAL06646 | *Citrus maxima* | flavonoid 1 – 2 rhamnosyltransferase |
| IpA3G2GT | BAD95882 | *Ipomoea purpurea* | anthocyanidin 3-glucoside 2′-O-glucosyltransferase |
| AtF5GT | AAM91686 | *Arabidopsis thaliana* | flavonoid 5-O-glucosyltransferase |
| AtF3GT | AAM91139 | *Arabidopsis thaliana* | flavonoid 3-O-rhamnosyltransferase |
| NtGT3 | BAB88934 | *Nicotiana tabacum* | glucosyltransferase |
| UGT73E5 | AB360611 | *Lycium barbarum* | |
| UGT73A10 | AB360612 | *Lycium barbarum* | glucosyltransferase |
| UGT75A2 | AB360613 | *Lycium barbarum* | |
| UGT73Q1 | AB360625 | *Lycium barbaru* | glucosyltransferase |

eight other UFGTs in cluster II. At their C-terminal ends, the ApUFGTs and the other UGTs grouped in the same cluster all contain the conserved PSPG domain that has been proposed to be a nucleoside-diphosphate-sugar binding site (figure 3).

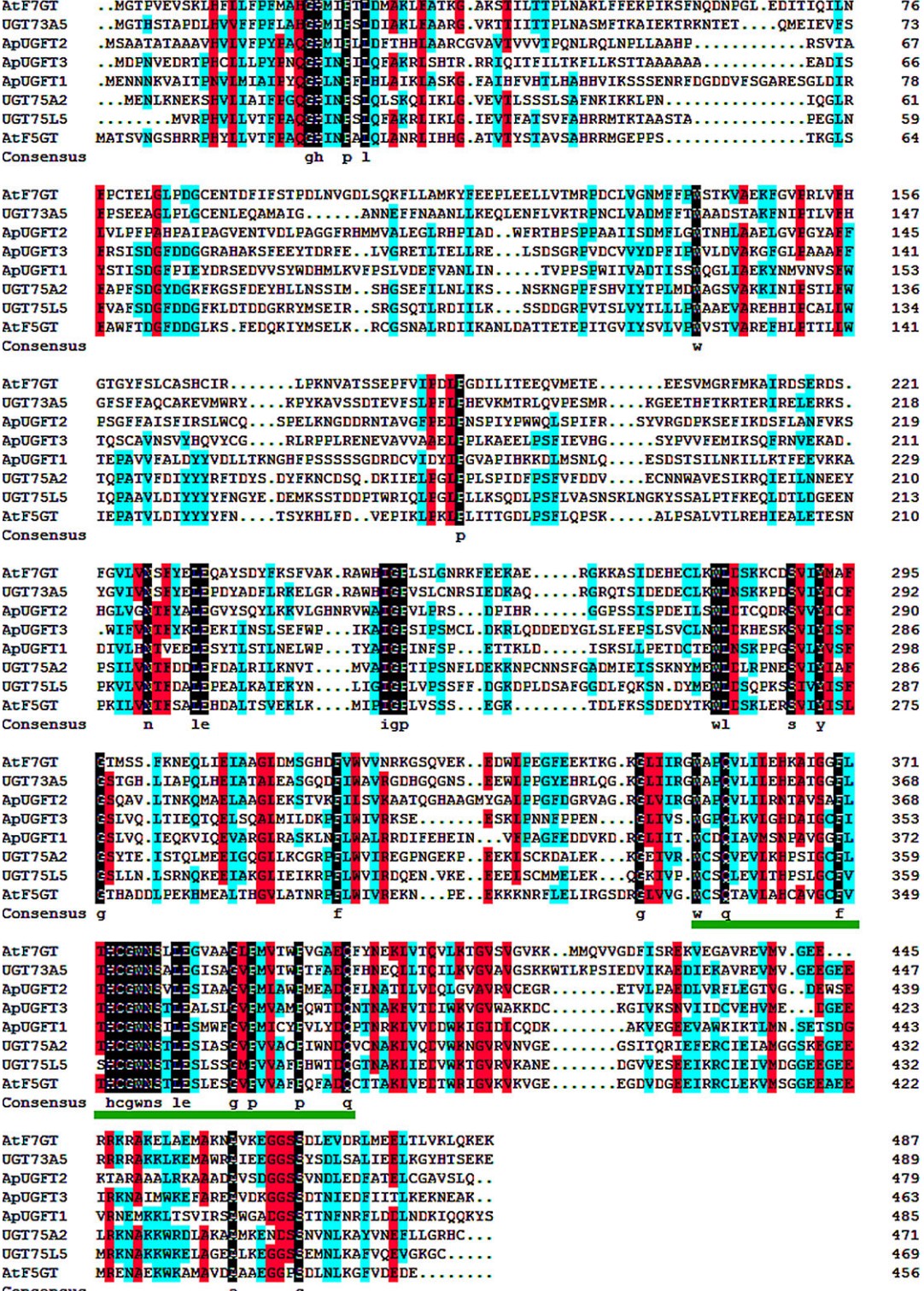

**Figure 3.** The amino acid sequence alignment of ApUFGTs and other plant UGTs. The multiple alignment was performed using DNAMAN software. The identified UGTs in multiple alignment are AtF7GT (AAL90934), UGT73A5 (CAB56231), UGT75A2 (AB360613), GT75L5 (AB360620) and AtF5GT (AAM91686). The green line indicates the conserved region of PSPG motif.

## 3.3. Study on the catalytic promiscuity of ApUFGTs

Several drug-like compounds with different types of structures were selected as substrates, including flavones (**1–6**), a flavonol (**7**), a flavanone (**8**), isoflavones (**9** and **10**), a dihydrochalcone (**11**), a coumarin (**14**) and other small molecular aromatic compounds with −OH and −NH$_2$ groups (**12** and **13**) (figure 4b).

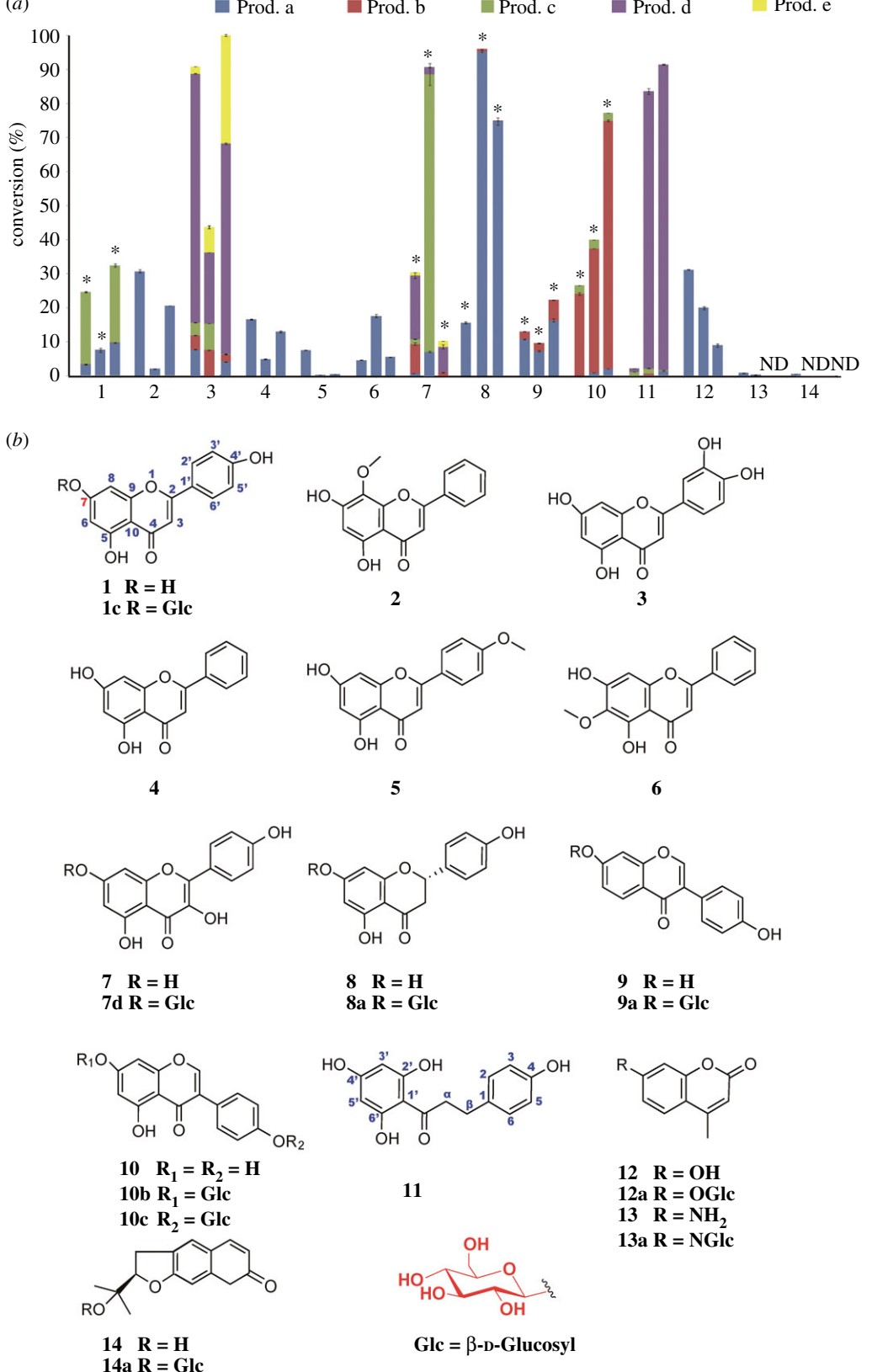

**Figure 4.** Exploring the catalytic promiscuity of the recombinant ApUFGTs. (*a*) Per cent conversion of glycosylated products catalysed by the ApUFGTs. The colour in the bar graphs (Prod. a, Prod. b, Prod. c, Prod. d and Prod. e) represent different ratios of diverse glycosylated products in the total product yield of each compound. Error bars used in the figure indicate $\pm$ s.d.s. The asterisks (*) represent the glucosylated products which were confirmed to be 7-*O*-glucosides by authentic standards. N.D. means no products detected. (*b*) Structures of the library members and corresponding glucosylated products.

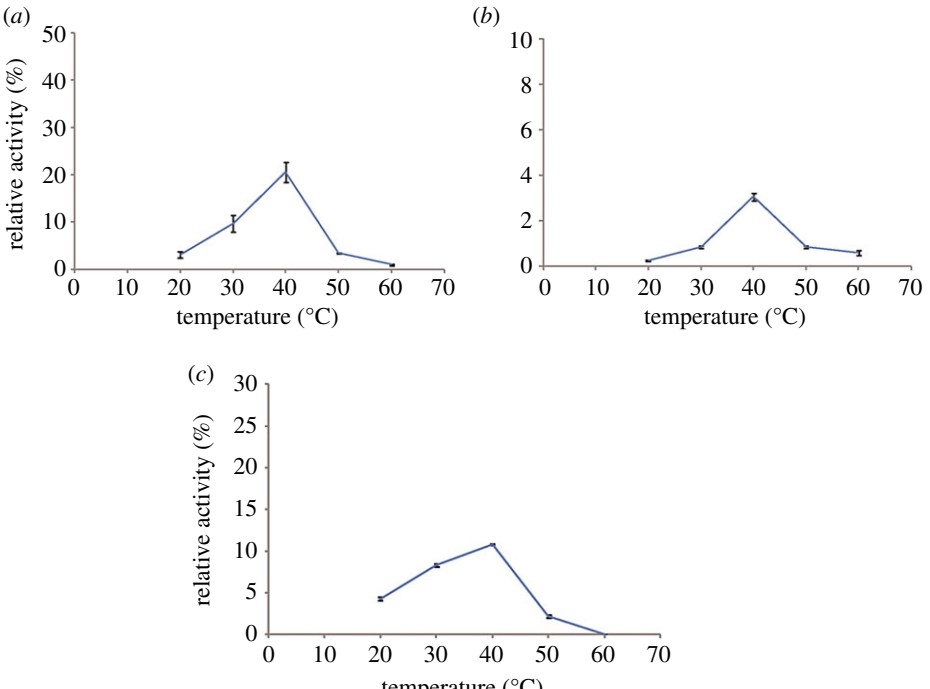

**Figure 5.** Effects of temperature on enzyme activity of ApUGT1 (*a*), ApUGT2 (*b*) and ApUGT3 (*c*).

The scopes of the substrates tolerated by the three recombinant ApUFGTs were systematically studied. For the same substrate, the types and conversion rates of the glycosylation products catalysed by the three glycosyltransferases are different (figure 4*a*), illustrating the diversity of plant secondary metabolic glycosyltransferases in *A. paniculata*. For the flavonoids in *A. paniculata* (**1**, **2** and **3**), the three ApUFGTs all exhibit glycosylation activity and can glycosylate multiple types of hydroxyl groups on certain substrates (**1** and **3**); there were at least two products in the glycosylation reaction of the two substrates. For certain flavonoids (**2**, **4**, **5** and **6**), the three ApUFGTs all exhibit strong positional selectivity, resulting in only one glycosylation product. Other flavonoids can be converted into different multi-site glycosylated products (**7–11**). Interestingly, for a non-natural (synthetic) substrate (**13**), ApUFGT1 and ApUFGT2 can catalyse the formation of *N*-glycoside bonds, highlighting the potential of these enzymes as multifunctional glycosylation tools.

## 3.4. UPLC-Q-TOF-MS confirmation of glycosylation products

The structures of the glycosylated products were verified using UPLC-Q-TOF-MS analysis by comparing the retention time (*t*), UV (λmax) and parent ions ([M + H]$^+$) of the glycosylated products with the corresponding standards (electronic supplementary material, figures S1–S15). Product peak **1c** was confirmed as apigetrin, namely, apigenin 7-*O*-glucoside. Product peak **7d** was identified as populnin, namely, kaempferol-7-*O*-β-D-glucopyranoside. Product peak **8a** was identified as prunin, naringenin-7-*O*-β-D-glucoside. Product peak **9a** was identified as daidzin, namely, daidzein 7-*O*-β-D-glucopyranoside. Product peak **10b** was identified as genistin, namely, genistein 7-*O*-glucoside.

## 3.5. Biochemical properties of ApUFGTs

Temperature and pH are two of the most important factors affecting enzymatic activity. The effects of different reaction temperatures (20–60°C) on the glycosylation of wogonin catalysed by ApUFGTs were investigated. The results showed that the three ApUFGTs all exhibited the highest enzymatic activity at 40°C. When the reaction temperature exceeded 40°C, the enzymatic activity decreased rapidly with increasing temperature, while the enzymatic activity remained low when the reaction temperature was below 30°C (figure 5). Therefore, the optimum temperatures of the ApUFGTs were all approximately 40°C. In the range of pH 7–8, the ApUFGTs all had higher enzymatic activities, but

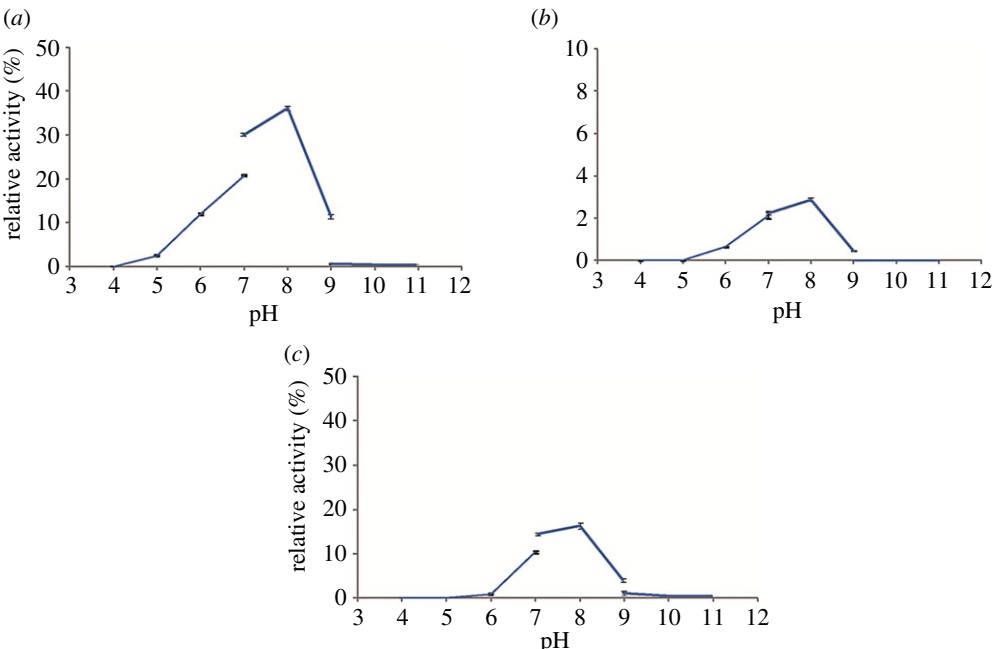

**Figure 6.** Effects of pH on enzyme activity of ApUGT1 (*a*), ApUGT2 (*b*) and ApUGT3 (*c*).

when the pH of the reaction system was below 6 or above 8, the activity decreased significantly (figure 6). Therefore, the optimum pH values for ApUFGTs were all approximately 8.0.

# 4. Discussion

Glycosyltransferases bear considerable importance owing to the fact that glycan moiety forms a necessary element of the plant secondary metabolisms, and can alleviate these disadvantages of chemical glycosylation [30]. With the progress in next-generation sequencing technologies and the reduction in the cost of sequencing, transcriptome analyses have become an important method for identifying the genes that participate in the biosynthesis of natural products, which can provide genetic information, gene expression levels and the basis for subsequent screens of candidate genes [52]. To identify the enzyme responsible for the glycosylation of flavonoids in *A. paniculata*, time-coursed transcriptome sequencing with MeJA treatment was performed, which may be a powerful tool for the further characterization of UGTs or other genes involved in secondary metabolisms.

Although significant progress has been made recently in the identification of putative UGT genes of many plant species, reports that documented the characterization of UGT family proteins with catalytic promiscuity remain relatively small. Here, we report a total of three UGTs from the leaves of *A. paniculata*, which exhibited broad substrate tolerance towards multiple flavonoids and could glycosylate various hydroxyl sites on flavonoids *in vitro* to preferentially form 7-*O*-glucoside products. The phylogenetic and sequence analysis of ApUFGTs reflect the diversity of glycosyltransferases in the same plant. With the continuous mining of glycosyltransferases in *A. paniculata*, novel enzymes with glycosylation activity will be identified, and the biosynthetic pathway of active components in *A. paniculata* will also be elucidated.

# 5. Conclusion

Using transcriptome sequencing, we identified and characterized three glycosyltransferases from *A. paniculata* (ApUFGTs), all of which could glycosylate flavonoids with various structures and preferentially glycosylate the 7-OH of their A ring. These enzymes also exhibited catalytic promiscuity in the glycosylation of different hydroxyl groups on flavonoids. In addition, ApUGT1 and ApUGT2 were capable of catalysing *O*-, and *N*-glycosidic bond formation. Biochemical properties and phylogenetic analysis of ApUFGTs were also investigated. These three glycosyltransferases could be effective enzymatic tools for the synthesis of flavonoid glycosides with different types of

structures. This study not only holds considerable promise for the resource development of *A. paniculata*, but also facilitates further enzyme engineering in drug design and the discovery of new active leading compounds.

Data accessibility. The datasets supporting this article have been uploaded as part of the electronic supplementary material.

Authors' contributions. Y.L., L.-Q.H and W.G. conceived and designed the research. Y.L. and X.-L.L. performed the experiments. C.-J.-S.L., R.-S.W. and L.-P.K analysed the data. T.M. and Z.-H.Z. participated in the research. Y.L., W.G. and L.-Q.H. wrote the paper with contributions from all the authors. All authors gave final approval for publication.

Competing interests. There are no conflicts to declare.

Funding. This work was supported by National Natural Science Foundation of China (81891010, 81891013, 81673547), the Key project at central government level: the ability establishment of sustainable use for valuable Chinese medicine resources (grant no. 2060302), and the Support Project of High-level Teachers in Beijing Municipal Universities in the Period of 13th 5-year Plan (CIT&TCD20170324) and National Program for Special Support of Eminent Professionals.

Acknowledgements. We thank Dr Xiaohong Zhang (Karmanos Cancer Institute, USA) for providing HIS-MBP-pET28a plasmid.

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
