## [Reviewer comments · Royal Society Open Science]

Review History

RSOS-190150.R0 (Original submission)

Review form: Reviewer 1 (Qingyu Wu)

Is the manuscript scientifically sound in its present form?

Yes

Are the interpretations and conclusions justified by the results?

Yes

Is the language acceptable?

Yes

Is it clear how to access all supporting data?

Not Applicable

Do you have any ethical concerns with this paper?

No

Have you any concerns about statistical analyses in this paper?

No

Recommendation?

Accept with minor revision (please list in comments)

Comments to the Author(s)

Comments:

Overall, the manuscript provides the discovery and characterization of three glycosyltransferases from *Andrographis paniculata* involved in the biosynthesis of the plant flavonoids. These candidate UFGTs were cloned, heterologously expressed, and purified from *E. coli*. It is clear that the authors discovered three glycosyltransferases with broad substrate specificities. The data that the authors present, is highly novel, particularly the identification and characterization of the flavonoids glycosyltransferases from *Andrographis paniculata*. These enzymes can be used for modifying flavonoid-like compounds to improve the biological activity, and can be potentially used in synthetic biology. The strength of this manuscript is its novelty and the extensive substrate specificity studies. Overall, this is nice work on a challenging system. However, there are grammatical and spelling errors in the manuscript. The manuscript will need to be clarified prior to publication. The errors did not prevent the material from being understood, but without correction, it would just be sloppy presentation. Once addressed, I think the manuscript is suitable for publication.

For example, the summary "and performed multi-site glycosidation towards the substrate molecules. Especially, they can preferentially act on the 7-OH position of the A ring" should revise to "the enzymes identified exhibit multi-site glycosidation towards the substrate molecules. Specifically, they can preferentially act on the 7-OH position of the A ring. "; Line 34 of page 2 change to "7-OH group of flavonoids as well as catalysing the glycosylation of flavones";

Line 18 of page 4, "and they all" change to "both of which";

Line 37 of page 4, change to "multiple types of hydroxyl groups", also line 38 of page 4, change to "For certain flavonoids".

As for the conclusions, I would write something like "We identified and isolated three novel substrate-promiscuous glycosyltransferases from *Andrographis paniculata* (ApUFGTs), all of which could glycosylate flavonoids with various structures and preferentially glycosylate 7-OH of the A ring. These enzymes also exhibited multiple functions involving the glycosylation of different hydroxyl groups on flavonoids" and "hold considerable promise for further resource development".

Also, latin names should be in italics in the whole manuscript.

Further minor points on Figure 4: Use "*" or other symbols to represent the glucosylated products which were determined to be 7-O-glucosides compared with the standard.

Review form: Reviewer 2**Is the manuscript scientifically sound in its present form?**

Yes

Are the interpretations and conclusions justified by the results?

Yes

Is the language acceptable?

Yes

Is it clear how to access all supporting data?

Yes

Do you have any ethical concerns with this paper?

No

Have you any concerns about statistical analyses in this paper?

No

Recommendation?

Accept with minor revision (please list in comments)

Comments to the Author(s)

In this manuscript, the authors report three novel flavonoid glycosyltransferases from the leaves of *Andrographis paniculata*. Substrate spectrum investigation demonstrated that the three glycosyltransferases exhibited substrate promiscuity with other type compounds as substrates except for flavonoids. These results are very interesting, and the manuscript is recommended to be accepted after minor revision.

Suggestion:

The biochemical properties of these new glycosyltransferases should be investigated.

Review form: Reviewer 3

Is the manuscript scientifically sound in its present form?

Yes

Are the interpretations and conclusions justified by the results?

Yes

Is the language acceptable?

No

Is it clear how to access all supporting data?

Yes

Do you have any ethical concerns with this paper?

No

Have you any concerns about statistical analyses in this paper?

No

Recommendation?

Accept with minor revision (please list in comments)

Comments to the Author(s)

Three flavonoid glycosyltransferases were isolated from *Andrographis paniculate* and their catalytic promiscuity towards flavonoids were verified. There are some comments should be addressed.

1, Enzyme Assays

"m/z" should be italic.

2, cDNA cloning of ApUFGTs

"*Olea europaea*" should be italic.

"O" in "3-O-glucosyltransferase" should be italic, the same below.

3, More attention should be paid to a clear descriptive legend, and this should be revised for all figures including supporting figures.

4, Where should Figure S1 be inserted in the text?

5, Check the structures, substitution positions and stereochemistry of all substrates. Also, flavonoids should be numbered.

6, Conclusion

"Novel" should be deleted.

7, A linguistic editing is required for correct grammar throughout the article.

8, Where are Tables S1 and S2? Authors may have miswritten Tables S1 and S2 into Tables 1 and 2.

Decision letter (RSOS-190150.R0)

03-Apr-2019

Dear Dr li,

The editors assigned to your paper ("Functional characterization of three novel flavonoid glycosyltransferases from *Andrographis paniculata*") have now received comments from reviewers. We would like you to revise your paper in accordance with the referee and Associate Editor suggestions which can be found below (not including confidential reports to the Editor). Please note this decision does not guarantee eventual acceptance.

Please submit a copy of your revised paper before 26-Apr-2019. Please note that the revision deadline will expire at 00.00am on this date. If we do not hear from you within this time then it will be assumed that the paper has been withdrawn. In exceptional circumstances, extensions may be possible if agreed with the Editorial Office in advance. We do not allow multiple rounds of revision so we urge you to make every effort to fully address all of the comments at this stage. If deemed necessary by the Editors, your manuscript will be sent back to one or more of the original reviewers for assessment. If the original reviewers are not available, we may invite new reviewers.

The reviewers have suggested that the quality of English needs to be improved. A number of language polishing services are available for authors whose first language is not English. <https://royalsociety.org/journals/authors/language-polishing/>

Authors whose papers are returned on language grounds must provide evidence that a professional language editing service or a native speaker of English have assisted in preparing a

revised manuscript. Evidence such as a certificate of editing or a signed letter from a native speaker of English would be acceptable.

- Data accessibility

If you wish to submit your supporting data or code to Dryad (<http://datadryad.org/>), or modify your current submission to dryad, please use the following link:
<http://datadryad.org/submit?journalID=RSOS&manu=RSOS-190150>

- Competing interests

- Authors' contributions

- Acknowledgements

- Funding statement

Kind regards,

Andrew Dunn

Comments to Author:

Reviewers' Comments to Author:

Reviewer: 1

Comments to the Author(s)

Comments:

Overall, the manuscript provides the discovery and characterization of three glycosyltransferases from *Andrographis paniculata* involved in the biosynthesis of the plant flavonoids. These candidate UFGTs were cloned, heterologously expressed, and purified from *E. coli*. It is clear that the authors discovered three glycosyltransferases with broad substrate specificities. The data that the authors present, is highly novel, particularly the identification and characterization of the flavonoid glycosyltransferases from *Andrographis paniculata*. These enzymes can be used for modifying flavonoid-like compounds to improve the biological activity, and can be potentially used in synthetic biology. The strength of this manuscript is its novelty and the extensive substrate specificity studies. Overall, this is nice work on a challenging system. However, there are grammatical and spelling errors in the manuscript. The manuscript will need to be clarified prior to publication. The errors did not prevent the material from being understood, but without correction, it would just be sloppy presentation. Once addressed, I think the manuscript is suitable for publication.

For example, the summary "and performed multi-site glycosidation towards the substrate molecules. Especially, they can preferentially act on the 7-OH position of the A ring" should revise to "the enzymes identified exhibit multi-site glycosidation towards the substrate molecules. Specifically, they can preferentially act on the 7-OH position of the A ring.";

Line 34 of page 2 change to “7-OH group of flavonoids as well as catalysing the glycosylation of flavones”;

Line 18 of page 4, “and they all” change to “both of which”;

Line 37 of page 4, change to “multiple types of hydroxyl groups”, also line 38 of page 4, change to “For certain flavonoids”.

As for the conclusions, I would write something like “We identified and isolated three novel substrate-promiscuous glycosyltransferases from *Andrographis paniculata* (ApUFGTs), all of which could glycosylate flavonoids with various structures and preferentially glycosylate 7-OH of the A ring. These enzymes also exhibited multiple functions involving the glycosylation of different hydroxyl groups on flavonoids” and “hold considerable promise for further resource development”.

Also, latin names should be in italics in the whole manuscript.

Further minor points on Figure 4: Use “*” or other symbols to represent the glucosylated products which were determined to be 7-O-glucosides compared with the standard.

Reviewer: 2

Comments to the Author(s)

In this manuscript, the authors report three novel flavonoid glycosyltransferases from the leaves of *Andrographis paniculata*. Substrate spectrum investigation demonstrated that the three glycosyltransferases exhibited substrate promiscuity with other type compounds as substrates except for flavonoids. These results are very interesting, and the manuscript is recommended to be accepted after minor revision.

Suggestion:

The biochemical properties of these new glycosyltransferases should be investigated.

Reviewer: 3

Comments to the Author(s)

Three flavonoid glycosyltransferases were isolated from *Andrographis paniculata* and their catalytic promiscuity towards flavonoids were verified. There are some comments should be addressed.

1, Enzyme Assays

“m/z” should be italic.

2, cDNA cloning of ApUFGTs

“*Olea europaea*” should be italic.

“O” in “3-O-glycosyltransferase” should be italic, the same below.

3, More attention should be paid to a clear descriptive legend, and this should be revised for all figures including supporting figures.

4, Where should Figure S1 be inserted in the text?

5, Check the structures, substitution positions and stereochemistry of all substrates. Also, flavonoids should be numbered.

6, Conclusion

“Novel” should be deleted.

7, A linguistic editing is required for correct grammar throughout the article.

8, Where are Tables S1 and S2? Authors may have miswritten Tables S1 and S2 into Tables 1 and 2.

Author's Response to Decision Letter for (RSOS-190150.R0)

See Appendix A.

RSOS-190150.R1 (Revision)

Review form: Reviewer 1 (Qingyu Wu)

Is the manuscript scientifically sound in its present form?

Yes

Are the interpretations and conclusions justified by the results?

Yes

Is the language acceptable?

Yes

Is it clear how to access all supporting data?

Yes

Do you have any ethical concerns with this paper?

No

Have you any concerns about statistical analyses in this paper?

No

Recommendation?

Accept as is

Comments to the Author(s)

The authors addressed all my concerns.

Review form: Reviewer 2

Is the manuscript scientifically sound in its present form?

Yes

Are the interpretations and conclusions justified by the results?

Yes

Is the language acceptable?

Yes

Is it clear how to access all supporting data?

Yes

Do you have any ethical concerns with this paper?

No

Have you any concerns about statistical analyses in this paper?

No

Recommendation?

Accept as is

Comments to the Author(s)

The authors addressed the issues well, and the manuscript can be accepted in the present version.

Review form: Reviewer 3

Is the manuscript scientifically sound in its present form?

Yes

Are the interpretations and conclusions justified by the results?

Yes

Is the language acceptable?

Yes

Is it clear how to access all supporting data?

Yes

Do you have any ethical concerns with this paper?

No

Have you any concerns about statistical analyses in this paper?

No

Recommendation?

Accept as is

Comments to the Author(s)

These authours accordingly revised all these comments.

Decision letter (RSOS-190150.R1)

17-May-2019

Dear Dr li,

I am pleased to inform you that your manuscript entitled "Functional characterization of three

flavonoid glycosyltransferases from *Andrographis paniculata*" is now accepted for publication in Royal Society Open Science.

Reviewer comments to Author:

Reviewer: 1

Comments to the Author(s)

The authors addressed all my concerns.

Reviewer: 2

Comments to the Author(s)

The authors addressed the issues well, and the manuscript can be accepted in the present version.

Reviewer: 3

Comments to the Author(s)

These authors accordingly revised all these comments.

Follow Royal Society Publishing on Twitter: [@RSocPublishing](https://twitter.com/RSocPublishing)

Appendix A

Dear Editor,

We appreciate for your efforts in processing our submission and the detailed comments and suggestions from you and the reviewers. According to the suggestions, we have improved the English and revised the manuscript, we also used the editing service to enable the manuscript to make a better impression on the readers, please see the certificate of editing below. The responses are listed below. I wish that the current manuscript could meet the requirements of *Royal Society Open Science* in language and other recommendations for authors. If you have any question about this paper, just please contact me. Thank you very much.

Yours sincerely,

Yuan Li, PhD

Shandong University of Traditional Chinese Medicine, Jinan, 250355, P. R. China.

National Resource Center for Chinese Materia Medica, China Academy of Chinese Medical Sciences, Beijing 100700, P. R. China.

Tel: +86 10 8404 4340, Fax: +86 10 84027175

E-mail: llyyly@163.com

AMERICAN JOURNAL EXPERTS

EDITORIAL CERTIFICATE

This document certifies that the manuscript listed below was edited for proper English language, grammar, punctuation, spelling, and overall style by one or more of the highly qualified native English speaking editors at American Journal Experts.

Manuscript title:

Functional Characterization of three flavonoid glycosyltransferases in *Andrographis paniculata*

Authors:

Yuan Li, Xin-Lin Li, Chang-Jiang-Sheng Lai, Rui-Shan Wang, Li-Ping Kang, Ting Ma, Zhen-Hua Zhao, Wei Gao ,and Lu-Qi Huang

Date Issued:

April 16, 2019

Certificate Verification Key:

7184-943E-9696-D48A-7ABP

This certificate may be verified at www.aje.com/certificate. This document certifies that the manuscript listed above was edited for proper English language, grammar, punctuation, spelling, and overall style by one or more of the highly qualified native English speaking editors at American Journal Experts. Neither the research content nor the authors' intentions were altered in any way during the editing process. Documents receiving this certification should be English-ready for publication; however, the author has the ability to accept or reject our suggestions and changes. To verify the final AJE edited version, please visit our verification page. If you have any questions or concerns about this edited document, please contact American Journal Experts at support@aje.com.

American Journal Experts provides a range of editing, translation and manuscript services for researchers and publishers around the world. Our top-quality PhD editors are all native English speakers from America's top universities. Our editors come from nearly every research field and possess the highest qualifications to edit research manuscripts written by non-native English speakers. For more information about our company, services and partner discounts, please visit www.aje.com.

Responses to the comments of Reviewer 1

Comments:

1. For example, the summary “and performed multi-site glycosidation towards the substrate molecules. Especially, they can preferentially act on the 7-OH position of the A ring” should revise to “the enzymes identified exhibit multi-site glycosidation towards the substrate molecules. Specifically, they can preferentially act on the 7-OH position of the A ring.”;

Line 34 of page 2 change to “7-OH group of flavonoids as well as catalysing the glycosylation of flavones”;

Line 18 of page 4, “and they all” change to “both of which”;

Line 37 of page 4, change to “multiple types of hydroxyl groups”, also line 38 of page 4, change to “For certain flavonoids”.

As for the conclusions, I would write something like “We identified and isolated three novel substrate-promiscuous glycosyltransferases from *Andrographis paniculata* (ApUFGTs), all of which could glycosylate flavonoids with various structures and preferentially glycosylate 7-OH of the A ring. These enzymes also exhibited multiple functions involving the glycosylation of different hydroxyl groups on flavonoids” and “hold considerable promise for further resource development”.

Response 1: Thanks a lot for the reviewer’s notice. We have corrected the mentioned errors, including the summary, line 34 of page 2, line 18 of page 4, line 37 of page 4, line 38 of page 4 and the conclusions. Additionally, we have polished the manuscript thoroughly.

2. Also, latin names should be in italics in the whole manuscript.

Response 2: Thank you for your helpful advice. We have italicized all the latin names in the whole manuscript.

3. Further minor points on Figure 4: Use “*” or other symbols to represent the glucosylated products which were determined to be 7-O-glucosides compared with the standard.

Response 3: Thank you for catching this oversight. In the revised manuscript, we have used “*” to represent the glucosylated products which were determined to be 7-O-glucosides compared with the standard in Figure 4.

Responses to the comments of Reviewer 2

The biochemical properties of these new glycosyltransferases should be investigated.

Response 3:

Thank you very much for your helpful advice. As suggested, we have investigated the biochemical properties of these glycosyltransferases. The results showed that the three ApUFGTs all exhibited the highest enzymatic activity at 40°C, and the optimum pH values for ApUFGTs were all approximately 8.0.

Responses to the comments of Reviewer 3

1, Enzyme Assays

“m/z” should be italic.

2, cDNA cloning of ApUFGTs

“Olea europaea” should be italic.

“O” in “3-O-glucosyltransferase” should be italic, the same below.

Response 1: We thank the reviewer for this astute observation. We have italicized the “m/z”, “Olea europaea” and “O” in the whole manuscript.

3, More attention should be paid to a clear descriptive legend, and this should be revised for all figures including supporting figures.

Response 2: Thank you for this helpful suggestion. In the revised manuscript, we have described all figures more clearly, including supporting figures.

4, Where should Figure S1 be inserted in the text?

Response 3: Thank you for catching this oversight. We inserted Figure S1 in the text of “UPLC-Q/TOF-MS Confirmation of Glycosylation Products”.

5, Check the structures, substitution positions and stereochemistry of all substrates. Also, flavonoids should be numbered.

Response 4: We thank the reviewer for this suggestion, which we have incorporated into the revised manuscript.

6, Conclusion

“Novel” should be deleted.

Response 5: We thank the reviewer for this correction, we have deleted the “Novel” in the manuscript .

7, A linguistic editing is required for correct grammar throughout the article.

Response 6: Thank you for the kind suggestion. We have extensively revised the manuscript, and carefully proofread the text in an attempt to catch all typographical errors and grammar. We also used the editing service to enable the manuscript to make a better impression on the readers.

8, Where are Tables S1 and S2? Authors may have miswritten Tables S1 and S2 into Tables 1 and 2.

Response 7: Thank you for your helpful advice. We have corrected these mistakes.